# Orthorexia Nervosa, Eating Disorders, and Obsessive-Compulsive Disorder: A Selective Review of the Last Seven Years

**DOI:** 10.3390/jcm11206134

**Published:** 2022-10-18

**Authors:** Maria Pontillo, Valeria Zanna, Francesco Demaria, Roberto Averna, Cristina Di Vincenzo, Margherita De Biase, Michelangelo Di Luzio, Benedetta Foti, Maria Cristina Tata, Stefano Vicari

**Affiliations:** 1Child and Adolescent Neuropsychiatry Unit, Department of Neuroscience, IRCCS Children Hospital Bambino Gesù, 00165 Rome, Italy; 2Department of Life Sciences and Public Health, Catholic University of the Sacred Heart, 00168 Rome, Italy

**Keywords:** orthorexia nervosa, eating disorders, obsessive-compulsive disorders, ORTO-15, anorexia nervosa, bulimia nervosa

## Abstract

Orthorexia nervosa (ON) is defined as an exaggerated, obsessive, pathological fixation on healthy food, healthy eating, or health-conscious eating behaviors. In the literature, there is an ongoing debate over whether ON should be considered simply a lifestyle phenomenon or a psychiatric disorder. In this vein, ON seems to share psychopathological characteristics with both eating disorders (EDs) and obsessive-compulsive disorder (OCD). However, there are insufficient data to reconcile the debate. The present study aimed at consolidating evidence on the clinical significance of ON and its relationship with EDs and OCD. A selective review of the literature published between January 2015 and March 2022 was conducted, following the Preferred Reporting Items for Systematic Reviews and Meta-Analyses (PRISMA) guidelines. Ten studies were included. Some of these studies suggested that ON might follow a full-syndrome DSM-5 ED. Other studies proposed that ON and DSM-5 EDs may co-occur. Finally, only two studies suggested a relationship between ON and OCD. To date, the clinical significance of ON and its relationship with EDs and other DSM-5 psychiatric disorders (e.g., OCD) appears complicated and unclear. Future longitudinal research on the possible clinical course of ON is needed.

## 1. Introduction

In 1997, the physician Steven Bratman first described orthorexia nervosa (ON) as a condition characterized by a pathological obsession with healthy or “pure” eating and proper nutrition.

In everyday life, individuals with ON avoid eating with other people and spend significant time planning and preparing healthy meals, using “pure food”. Notably, they exclude foods from their diets that they consider impure because they have pesticides or artificial substances, and they worry in excess about the materials used in the food elaboration. This specific attention to food purity and healthiness becomes an obsession that significantly interferes with global life functioning and may also lead to physical impairments such as malnutrition and excessive weight loss [1,2,3]. At present, ON is not recognized as a psychopathological disorder in the DSM-5 or the ICD-10 [4]. For this reason, the disorder has no established and validated diagnostic criteria. Indeed, in the literature, there is an ongoing debate over whether ON should be considered simply a behavioral/lifestyle phenomenon or a psychiatric disorder. However, some studies have described specific clinical features of the disorder [5,6,7]. 

ON is typically characterized by a desire to enhance one’s health and prevent or treat disease through the consumption of healthy food and the elimination of foods perceived as impure and unhealthy; this generally results in severe food intake restriction and the elimination of specific food groups, such as meat, dairy, grain, cooked food, and non-seasonal produce [5,6]. Another important characteristic of ON is that the associated dietary regime is not part of a medically prescribed diet, and the research, analysis, and preparation of food may take significant time (i.e., often more than 3 h per day) [4,5]. In individuals with ON, such food behaviors become obsessive preoccupations that are accompanied by a pathological level of worry and stress concerning food [7]. However, healthy eating represents the first-line prevention for various diseases, and some studies have focused on orthorexia in specific clinical populations, for example, patients with diabetes mellitus [8]. 

Several instruments exist for the evaluation of ON, but the most widely used in the literature is the ORTO-15. The ORTO-15 is a self-report questionnaire [9] and made up of 15 multiple-choice items (e.g., When eating, do you pay attention to the calories of the food? Are you willing to spend more money to have healthier food? In the last three months, did the thought of food worry you). Total scores <35 are considered pathological, and low scores, more generally, are significantly associated with a high presence of orthorexic behaviors. According to Donini et al. [9], the overall accuracy of the ORTHO-15 test as a test for the diagnosis of ON was found to be 0.696 (95% CI: 0.585–0.807). However, different versions of the instrument (e.g., ORTO-14, ORTO-R) contain different numbers of items. Other instruments are also used for the evaluation of ON. For example, the Düsseldorf Orthorexie Scale (DOS) is a 10-item self-reported questionnaire and measures orthorexic eating behavior. A four-point Likert scale from “this applies to me” (four points) to “this does not apply to me” (one point) is used. Higher points indicate more pronounced orthorexic behavior. As a cut-off score to indicate the presence of ON, a score ≥30 is used. DOS has acceptable internal consistency (α = 0.84) [10]. Finally, the Revised Bratman’s Orthorexia Test [rBOT] [11] is an eight-item modified version of the Bratman Orthorexia Test [BOT] [5]. Scores are summed with a possible range of 0–16. The rBOT has good internal consistency (α = 0.85) [12]”.

Previous research [13,14,15] has revealed that the clinical characteristics of ON are similar to those of two psychopathological disorders: eating disorders (EDs) and obsessive-compulsive disorder (OCD). As concerns about the relationship between ON and Eds, Dell’Osso et al. [16] proposed that ON and Eds (e.g., anorexia nervosa) share abnormal eating attitudes with dieting behavior and poor insight. Regarding this, a previous study [17] showed that dieting behavior is among the factors that increase the risk for Eds. Other factors are thin ideal, body dissatisfaction, and unhealthy weight control behaviors. Finally, ON is likely to lead to many of the same medical complications as EDs. For example, a lack of essential nutrients caused by restrictive eating can result in malnutrition, anemia, or an abnormally slow heart rate [7]. Severe malnutrition may lead to gastrointestinal problems, hormonal imbalances, general fatigue, and a weakened immune system [18].

However, other studies [16] have suggested that, even if ON resembles EDs, individuals with ON are worried about the quality of food and the healthy preparation of meals, while individuals with EDs (in particular anorexia nervosa and bulimia nervosa) are mainly worried about the quantity of food intake and the number of calories ingested, in order to maintain a very low body weight [6,19]. While in ON, weight loss is a common consequence of the highly restrictive diet, the alimentary behaviors aim at achieving feelings of purity and good health [20]. Finally, Atchison and Zickgraf [21], in a recent systematic review, found that ON symptoms are related to the effort to lose or contain weight but not to body dissatisfaction or dysregulated eating. Therefore, according to these authors, ON may represent a distinct ED.

Research has also investigated the relationship between ON and OCD. Similar to patients with OCD, individuals with ON spend most of their time practicing strict and effortful procedures to choose, prepare, and eat healthy food. Indeed, orthorexic tendencies have been found in patients with OCD symptoms. In addition, some individuals with ON present recurrent, intrusive thoughts related to food, health, and contamination, which cause marked distress and anxiety. Most likely, these thoughts are similar to the obsessions experienced by OCD patients. Furthermore, the ritualistic processes of food preparation and consumption observed in individuals with ON resemble compulsions. Similar to the obsessions and compulsions of OCD patients, orthorexic thoughts and behaviors generate significant wasted time and limitations to social life resulting in high distress and worsening of quality of life [7]. 

Overall, there is limited systematic research on ON, its clinical significance, and its relationship with psychiatric disorders (e.g., EDs, OCD). The studies and reviews described so far have specifically considered the relationship between ON and eating disorders or that between ON and OCD.

Thus, the present review aims to summarize the literature over the last seven years on the clinical significance of ON and its relationship with both EDs and OCD. 

## 2. Methods

The present study comprised a selective review of the literature published between January 2015 and March 2022, according to the Preferred Reporting Items for Systematic Reviews and Meta-Analyses (PRISMA) guidelines.

### 2.1. Search Strategy

All included studies were obtained from a literature search of the electronic databases PubMed, CINAHL, PsycInfo, MedLine, and Cochrane Library. The algorithm used for the literature research was a combination of the terms: (orthorexia nervosa) AND (eating disorder AND obsessive-compulsive disorder AND mental disorders). We purposely added “mental disorders” as a keyword in order to balance the sensitivity and specificity of our search strategy.

Twelve articles were included. The last update of the search was in March 2022. 

### 2.2. Inclusion and Exclusion Criteria

The included studies investigated the clinical significance of ON and its relationship with EDs and OCD. Reviews, meta-analyses, comments, and letters were excluded. No language restrictions or study design restrictions were applied.

### 2.3. Selection Procedure, Data Extraction, and Data Management 

The reference lists of the most important articles of interest were examined. Data on efficacy, acceptability, and tolerability were extracted by three authors independently (i.e., S.V., M.P., V.Z.). Article selection was completed by other authors (i.e., B.F., C.D.V., M.C.T, M.D.B., M.D.L., R.A., F.D.). The search algorithm retrieved a total of 175 articles, of which 140 were excluded prior to screening. Of the 35 records screened, 10 referred to eligible studies, and the remaining 25 were excluded for the reasons listed in Table 1.

In terms of evidence-based medicine, the quality of the included studies was moderate. Figure 1 presents a detailed flow diagram of the study selection process.

## 3. Results

Ten studies on the clinical significance of ON and its relationship with DSM-5 EDs and OCD were analyzed. Due to the number and heterogeneity of the included studies, a narrative synthesis was conducted to describe, organize, explore, and interpret their findings while examining their methodological adequacy. Table 2 describes the methodologies and results of the studies.

### 3.1. Clinical significance of Orthorexia Nervosa 

Recently, Strahler et al. [46] examined if ON is merely a behavioral/lifestyle phenomenon or a mental disorder. ON symptoms (measured using the Düsseldorf Orthorexie Scale [DOS]) were detected in 27 of a total sample of 713 individuals (3.8%). These ON individuals reported lower well-being, lower life satisfaction, and higher stress compared to non-ON individuals. Regarding the overlap with other psychopathological eating and mental health problems, pathological eating (measured using the Eating Disorder Examination-Questionnaire 8 [EDE-Q8]) was detected in 21 of the ON subjects (77.8%). Moreover, anxiety and depressive symptoms (measured using the Patient Health Questionnaire-9 [PHQ-9]) were detected in 48.2% of the ON subjects. Finally, obsessive and compulsive behaviors (measured using the Yale Brown Obsessive-Compulsive Scale and Symptom Checklist [Y-BOCS]) were detected in 30% of the ON subjects.

Łucka et al. [47] examined the prevalence of ON in 864 adolescents and young adults (age range: 13–30 years). The results demonstrated that 27.78% of the sample presented a risk of developing orthorexia nervosa, according to the ORTO-15. Comparing the results of the ORTO-15 and the Eating Attitude Test (EAT-26), the authors found a significant association between EDs and ON (*p* < 0.001). A major risk of developing ON in adolescents aged 13–16 years was also found. Comparing the results of the ORTO-15 and the Maudsley Obsessional-Compulsive Inventory (MOCI), the authors found no significant relationship between the severity of obsessive-compulsive symptomatology and ON.

### 3.2. Orthorexia Nervosa and DSM-5 Eating Disorders

Barthels et al. [48] investigated orthorexic eating behaviors in an inpatient treatment sample of 42 female patients with anorexia nervosa. Orthorexic eating behaviors were assessed using the DOS. The anorexic group was divided into two subgroups: an AN group composed of 29 patients with low levels of orthorexic eating behaviors, and an ANO group composed of 13 patients with high levels of orthorexic eating behaviors. Both groups scored significantly higher in a drive for thinness compared to the matched control group of healthy females (AN vs. control: *p* = 0.001; ANO vs. control: *p* = 0.001). Finally, regarding the fulfillment of basic psychological needs, the ANO group scored significantly higher on the Basic Psychological Needs Scale [BPNS-E] competence (*p* < 0.017) and autonomy (*p* < 0.017) subscales relative to the AN group. 

Brytek-Matera et al. [49] assessed orthorexic behaviors, ED pathology, and attitudinal body image in a sample of 52 women diagnosed with EDs. The test battery included the ORTO-15, the EAT-26 to identify ED symptoms, and the Multidimensional Body-Self Relations Questionnaire (MBSRQ) to investigate body image perception. The first result was that 82.7% (*n* = 43) of subjects presented a strong preoccupation with healthy food intake. Additionally, ON was negatively predicted by eating pathology, weight concern, health orientation, and appearance orientation. In fact, eating pathology and orthorexic behaviors showed a negative association, whereby lower levels of eating pathology were associated with more frequent orthorexic behaviors.

Bartel et al. [12] investigated whether ON bears a stronger relationship with EDs or OCD in a sample of 512 young adults, using the Revised Bratman’s Orthorexia Test [rBOT], the Eating Disorder Examination Questionnaire (EDE-Q), the Obsessive-Compulsive Inventory-Revised (OCI-R), the Food Choice Questionnaire, and the Frost Multidimensional Perfectionism Scale (FMPS). The results showed that ON symptoms were more strongly linked to those of EDs than those of OCD. ON symptoms were related to body weight and shape concerns, and food selection tended to prioritize weight over health (*p* < 0.01). Furthermore, correlations of the FMPS total score with scores on the OCI-R and EDE-Q were significantly greater than those of the FMPS total score with that of the rBOT (*p* < 0.05). Similarly, the association between the FMPS total score with that of the OCI-R was significantly greater than the association between the FMPS total score with that of the EDE-Q (*p* < 0.01). 

Barnes and Caltabiano [50] investigated whether perfectionism, body image, attachment style, and self-esteem predicted ON in 220 participants through a self-administered online assessment consisting of five measures: the ORTO-15, the Multidimensional Perfectionism Scale (MPS), the Multidimensional Body-Self Relations Questionnaire-Appearance Scale (MBSRQ-AS), the Relationship Scales Questionnaire (RSQ), and Rosenberg’s Self-Esteem Scale (RSES). The correlation analysis revealed that higher orthorexic tendencies were significantly associated with higher perfectionism (i.e., self-oriented, others-oriented, socially prescribed), appearance orientation, overweight preoccupation, self-classified weight, and fearful and dismissing attachment styles. Additionally, the most important predictors of ON emerged as a history of an ED (*p* = 0.005), followed by appearance orientation (*p* = 0.002) and overweight preoccupation (*p* < 0.001). 

Novara et al. [51] examined whether ON may be related to and differentiated from OCD, EDs, perfectionism, anxiety, and depression in 302 individuals from the general population. The sample was divided into two groups, named “High EHQ” and “Low EHQ,” based on scores on the Eating Habits Questionnaire (EHQ-21). The results showed a correlation between ON and EDs and non-adaptive perfectionism constructs, which emerged independently of obsessive-compulsive symptoms. The same pattern was observed when comparing the High and Low EHQ groups. 

### 3.3. Orthorexia Nervosa and Obsessive-Compulsive Disorder

Vaccari et al. [52] analyzed the prevalence and intensity of ON symptoms (measured using the ORTO-15) in patients who had been diagnosed with obsessive-compulsive symptoms (measured using the OCI-R), in comparison to 42 subjects with anxiety or depression disorders and 236 subjects with no psychiatric morbidity. The main finding was that patients with OCD presented higher ON symptoms when compared to patients in the other groups (*p* = 0.0005). 

Yilmaz et al. [53] investigated the relationship between ON and obsessive-compulsive symptoms, eating attitudes, and sociodemographic features in a sample of 63 OCD patients, 63 healthy volunteers who regularly engaged in physical exercise, and 63 healthy volunteers who did not regularly engage in physical exercise. The results showed a statistically significant relationship between order-symmetry obsessions (measured using the Y-BOCS Symptom Checklist) and orthorexic tendencies in patients with OCD (measured using the ORTO-11) (*p* < 0.05). No relationship was found between the severity of obsessive-compulsive symptoms and ON in patients with OCD (*p* > 0.05). Additionally, orthorexic tendencies were higher in participants who regularly engaged in physical exercise and patients with OCD (*p* < 0.05). When the groups were compared according to scores on the ORTO-11, the orthorexic symptoms of participants who regularly engaged in physical exercise were higher than those of the OCD patients and the healthy individuals who did not regularly engage in physical exercise (*p* < 0001).

Rania et al. [54] described four clinical cases in which patients had symptoms suggestive of ON. The final sample was formed of four women with a prior psychiatric disorder other than ON who had been referred to an ED unit for the development of symptoms compatible with ON. All patients were interviewed by experienced psychiatrists and screened for psychiatric disorders according to the Structured Clinical Interview for DSM-5 and EDs through the Eating Disorder Examination. Dunn and Bratman’s criteria were used during the clinical interview to investigate ON symptoms. The ORTO-15 was administered to patients in support of the clinical diagnosis, considering a more restrictive cut-off (i.e., a pathological score < 35 instead of <40), as previously reported in other studies. The first patient was a 32-year-old woman who had been diagnosed with OCD at the age of 27. She had undergone drug therapy (i.e., sertraline) and cognitive-behavioral therapy for two years, had become worried about food choices, and developed egosyntonic obsessions and compulsions around healthy eating (ORTO-15 score = 34). The second patient was a 24-year-old woman who had been diagnosed with anorexia nervosa and then bulimia nervosa at the age of 20, before later developing worries and obsessions around healthy food (ORTO-15 score = 34). The third patient was a 45-year-old mother who, in her early anamnesis, reported symptoms of a possible illness anxiety disorder. At the time of the consultancy, she revealed some worries about the danger of food, which led to an obsession with “safe food” and evident malnutrition (ORTO-15 score = 32). The final patient was a 39-year-old woman who had been diagnosed with paranoid personality traits and a psychotic disorder not otherwise specified, for which she received antipsychotic treatment. For some years, she had been suffering from paranoid thoughts about food poisoning, which resulted in her qualitatively restricting her diet (ORTO-15 score = 33). 

**Table 2 jcm-11-06134-t002:** Methodologies and results of the investigated studies.

Study	Sample	Method(s)	Measures	Results
Barnes and Catalbiano [50]	Sample: 220 adults (180 university psychology students, 40 recruited from Facebook; 46 male, 154 female; age range: 17–62 years [M = 23.81, SD = 8.40])	Experimental study	ORTO-15; MPS; MBSRQ-AS; RSQ; RSES	History of an ED was the strongest predictor of ON; ORTO-15 scores were significantly correlated with perfectionism and fearful and dismissing attachment styles, but not with self-esteem
Novara et al. [51]	Sample: 302 university students in northern Italy. Total sample divided into two groups: “High EHQ” (*n* = 43; 22 male, 21 female, age range: 18–31 years [M = 20.60, SD = 2.44]) and “Low EHQ” (*n* = 259; 41.5% male, 58.5% female; age range: 18–49 years [M = 20.83, SD = 3.33])	Experimental study	EHQ-21; EDI-3; OCI-R; WDQ; PSWQ; MPS; BAI; BDI-II	Association between ON and perfectionism, anxious and depressive symptoms, and ED symptoms
Brytek-Matera et al. [49]	Sample: 52 women diagnosed with an ED (M_age_ = 22.81, SD = 3.80)	Experimental study	ORTO-15; MBSRQ; EAT-26	Lower level of eating pathology associated with more frequent orthorexic behaviors; higher level of eating pathology associated with less frequent orthorexic behaviors; ON negatively predicted by eating pathology, weight concern, health orientation, and appearance orientation
Barthels et al. [48]	Sample: 42 female anorexic patients with orthorexic eating behaviors (M_age_ = 21.17, SD = 6.88; M_BMI_ = 15.97, SD = 1.52 kg/m^2^) Control group: 30 females (M_age_ = 22.10, SD = 7.43 years; M_BMI_ = 21.83, SD = 2.75 kg/m^2^)	Experimental study	DOS; EDI-2; DKB-35; BPNS-E; MIHT	Orthorexic eating behaviors might represent coping mechanisms for patients with anorexic eating behaviors, and healthier ways of controlling food intake than focusing on low-calorie foods
Rania et al. [54]	Sample: 4 women with a prior psychiatric disorder (M_age_ = 35)	Case report	ORTO-15; SCID-5-CV	Some psychiatric conditions, across a diagnostic continuum, may lead to ON
Łucka et al. [47]	Sample: 864 adolescents and young adults from the general population (265 male, 599 female; age range: 13–30 years)	Experimental study	ORTO-15; EAT-26; MOCI	Significant association between EDs (EAT-26) and ON (ORTO-15); no significant relationship between the severity of obsessive-compulsive symptoms (MOCI) and orthorexia (ORTO-15)
Vaccari et al. [52]	OCD group: 50 patients Control group 1: 42 patients with a diagnosed anxiety or depressive disorder Control group 2:236 subjects from the general population	Multi-center, observational, controlled study	ORTO-15; ORTO-R; OCI-R	More ON symptoms among widowers relative to subjects with a partner and separated/divorced subjects; ON symptoms more prevalent in less educated subjects and those engaging in high-frequency physical activity; ORTO-R variation associated with a positive OCI-R score
Yilmaz et al. [53]	Sample: 189 individuals (79 outpatients with OCD, 68 healthy controls who regularly engaged in exercise, 69 healthy controls who did not regularly engage in exercise; age range: 18–65 years)	Experimental study	SCID-5/CV Y-BOCS; EAT-40; ORTO-11; HAS	Orthorexic symptoms increased in the E + HC group as eating attitude deteriorated; orthorexic tendencies were higher in subjects with order-symmetry obsessions than in those with no such obsessions
Bartel et al. [12]	Sample: 512 individuals recruited through social media, an undergraduate psychology pool, and the general student body of a university in Western Canada (89 male, 423 female; M_age_ = 24.5 years)	Experimental study	EDE-Q; rBOT; ORTO-15; OCI-R; FMPS; FCQ	Strong correlation between ON and EDE-Q (*r* = 0.63); correlation between ON and OCI-R total scales (*r* = 0.27); controlling for EDE-Q scores, only a small or no association between ON and OC symptoms (*r* = 0.08)
Strahler et al. [46]	Sample: 713 subjects recruited through public advertisements in local shops and mailing lists from universities in the broader Giessen/Marburg area (20.2% male, 79.8% female; age range: 18–75 years [M = 25])	Cross-sectional study	DOS; WHO-5; PSS-10; RS-13; WREQ; EDE-Q8; PHQ-9; HADS; AUDIT; Y-BOCS; GPPAQ	Strong correlation between ON and other mental disorders; ON no more prevalent than other forms of restrictive dieting and not associated with physical activity levels within a healthy lifestyle

MPS: Multidimensional Perfectionism Scale; MBSRQ-AS: Multidimensional Body-Self Relations Questionnaire-Appearance Scale; RSQ: Relationship Scales Questionnaire; RSES: Rosenberg Self-Esteem Scale; EHQ-21: Eating Habits Questionnaire; EDI-3: Eating Disorder Inventory-3; OCI-R: Obsessive Compulsive Inventory-Revised; BAI: Beck Anxiety Inventory; BDI-II: Beck Depression Inventory-Second Edition; WDQ: Worry Domains Questionnaire; PSWQ: Penn State Worry Questionnaire; MBSRQ: Multidimensional Body-Self Relations Questionnaire; EAT-26: Eating Attitude Test; DOS: Düsseldorfer Orthorexie Skala; EDI-2: Eating Disorder Inventory-2; DSB-35: DresdnerKorperbildfragebogen; BPNS-E: Basic Psychological Needs Scale; MIHT: Multidimensional Inventory of Hypochondriacal Traits; EDE-Q: Eating Disorder Questionnaire; SCID-5-CV: Structured Clinical Interview for DSM-5; MOCI: Maudsley Obsessive Compulsive Inventory; Y-BOCS: Yale Brown Obsessive-Compulsive Scale and Symptom Checklist; HAS: Hamilton Anxiety Scale; WHO-5: World Health Organization Well-Being Index; PSS-10: Perceived Stress Scale; WREQ: Weight-Related Eating Questionnaire; EDE-Q8: Eating Disorder Examination—Questionnaire; PHQ-9: Patient Health Questionnaire; HADS: Hospital Anxiety and Depression Scale; AUDIT: Alcohol Use Disorders Identification Test; GPPAQ: General Practice Physical Activity Questionnaire.

## 4. Discussion

The present selective review aimed at summarizing the recent findings on the clinical significance of ON and its relationship with EDs and other DSM-5 psychiatric disorders (e.g., OCD). 

### 4.1. Clinical Significance of Orthorexia Nervosa

Within the literature, there is an ongoing debate regarding two important clinical issues. The first consists of verifying whether ON is a behavioral/lifestyle phenomenon or a mental disorder. In this respect, Strahler et al. [46] found that, in a sample of 713 participants, 27 (3.8%, 21 women) presented with significant orthorexic eating behaviors. Additionally, in a sample of 864 adolescents and young adults (age range: 13–30 years), Łucka et al. [47] identified prevalent ON behaviors (measured using the ORTO-15) in 27.78% of participants. These numbers are aligned with previously published data [10,55]. Conclusions on the prevalence of ON should be drawn with caution, as the relevant studies have generated conflicting results depending on the analyzed population and the adopted diagnostic criteria. Despite this, all of the studies investigated as part of this review found that individuals with ON reported lower well-being, lower life satisfaction, higher stress, greater depressive symptoms, and lower global functioning relative to individuals without ON. On this basis, ON may be considered a condition with clinical significance and not simply a behavioral/lifestyle phenomenon. However, heterogeneous and culture-dependent definitions of what is considered normal eating behavior make longitudinal studies on the physical, psychological, and social consequences of ON complicated. Finally, differentiating ON from already existing psychiatric is essential for its classification as a “new” mental illness with associated clinically relevant distress or limitations in important areas of life (e.g., social isolation).

### 4.2. Orthorexia Nervosa and DSM-5 Eating Disorders

The second clinical issue about ON that is highly debated in the literature relates to whether or not ON should be classified as a distinct disorder, an ED, or a variant of another psychiatric disorder (e.g., OCD). As regards the clinical relationship between ON and DSM-5 EDs, Barnes and Caltabiano [50] showed that higher orthorexic tendencies significantly correlated with higher perfectionism (i.e., self-oriented, others-oriented, socially prescribed), overweight preoccupation, and appearance orientation. These findings are consistent with the results of other studies finding that perfectionism is implicated in the development and maintenance of EDs [56,57]. Also, greater appearance orientation and overweight preoccupation have been observed among anorexic and bulimic individuals [58]. Overall, this may suggest that orthorexia nervosa shares similarities with EDs with regard to perfectionism, overweight preoccupation, and appearance orientation. In addition, in Barnes and Caltabiano’s study [50], history of an ED, appearance orientation, and overweight preoccupation emerged as the most important predictors of ON. Overall, this may suggest that an exaggerated focus on appearance and a fear of becoming overweight might be underlying motivations for ON individuals’ preoccupation with a healthy diet rather than a fixation on health, per se.

Barthels et al. [48] showed that orthorexic eating behaviors might serve as coping mechanisms for patients with anorexic eating behaviors. Indeed, in this study, fulfillment of basic psychological needs with respect to autonomy and competence was higher amongst anorexic patients with pronounced orthorexic eating behaviors compared to those with low orthorexic eating behaviors. 

In anorexia, according to Fairburn [59], a feeling of self-control (especially with respect to food intake) is a crucial clinical feature. Therefore, orthorexic eating behaviors might represent healthier ways of controlling food intake, shifting the focus from low-calorie foods to healthy foods, and resulting in a greater variety of “allowed” foods for recovering anorexic individuals and a lower risk of losing weight. This proposal by Barthels et al. [48] is supported by Segura-Garcia et al.’s study [19], which found that orthorexic eating behaviors were associated with clinical improvement of disordered eating symptoms and a shift towards less severe forms of disturbed eating behaviors. Overall, since anorexia is difficult to treat and clinical changes need time, considering orthorexic eating behavior as a first step in a psychotherapeutic approach might be a promising goal of treatment.

Further research is needed to assess orthorexic-eating behaviors during treatment for anorexia, applying several follow-up measurements (i.e., not only self-report questionnaires) to investigate orthorexic and anorexic eating behaviors over time.

### 4.3. Orthorexia Nervosa and Obsessive-Compulsive Disorder

Regarding the clinical relationship between ON and OCD, previous studies [60,61,62] have shown that obsessive-compulsive symptoms can be present in orthorexic individuals. 

In our review, Novara et al. [51] focused on the relationship between worry’s content and ON clinical features. Specifically, ON is characterized by overthinking about healthy food and food preparation rituals. These aspects resemble the obsessions and compulsive behaviors typical of OCD. 

Similarly, Vaccari et al. [52] aimed at investigating the prevalence and intensity of ON symptoms in OCD patients, in comparison to subjects with anxiety and depressive disorders and healthy controls. They found higher ON symptoms in OCD patients. 

Overall, these results seem to support the idea that a correlation exists not only between ON and EDs but also between ON and OCD.

Despite this, we point out a crucial difference between ON and OCD. In individuals with ON, ritual food preparation is not aimed at reducing distress like in OCD compulsions.

Indeed, the repetitiveness and permanence of food-related thinking might suggest an excessive concern over the healthiness and preparation of food rather than an obsession followed by a compulsion to neutralize it. Finally, an individual with ON experiences his eating behavior as being in harmony or consistent with his ideal self-image. 

We propose that considering this egosyntonicity is a crucial target in improving the diagnostic definition of orthorexia and the implementation of effective psychotherapeutic treatments. Even if obsessive-compulsive characteristics may be associated with or mediate orthorexic behaviors, these two conditions could be treated with different approaches. 

For example, a psychoeducational intervention, focused on improving awareness of distress and the decrease in quality of life associated with orthorexic eating behaviors, could be useful in order to enhance compliance of individuals with ON with any other therapeutic intervention. 

## 5. Limitations

Some limitations should be considered in our review. Firstly, the included studies featured a variety of study designs and ON measures. In addition, the review included both general population and clinical population studies. This limits the interpretability of the findings.

## 6. Conclusions

The present selective review synthesized the recent literature on the prevalence and clinical significance of ON. The results suggest that ON should be considered a condition with clinical significance and not simply a behavioral/lifestyle phenomenon. In addition, while ON may be related to EDs and OCD, these relationships are not yet clear and should be clarified in future research. In particular, longitudinal studies should be conducted to collect data at the onset and during the clinical course of ON. This might improve our ability to determine whether ON precedes or follows the onset of EDs, OCD, or both.

This knowledge could help the better diagnostic definition of orthorexia and the development of adequate therapeutic interventions.

## Figures and Tables

**Figure 1 jcm-11-06134-f001:**
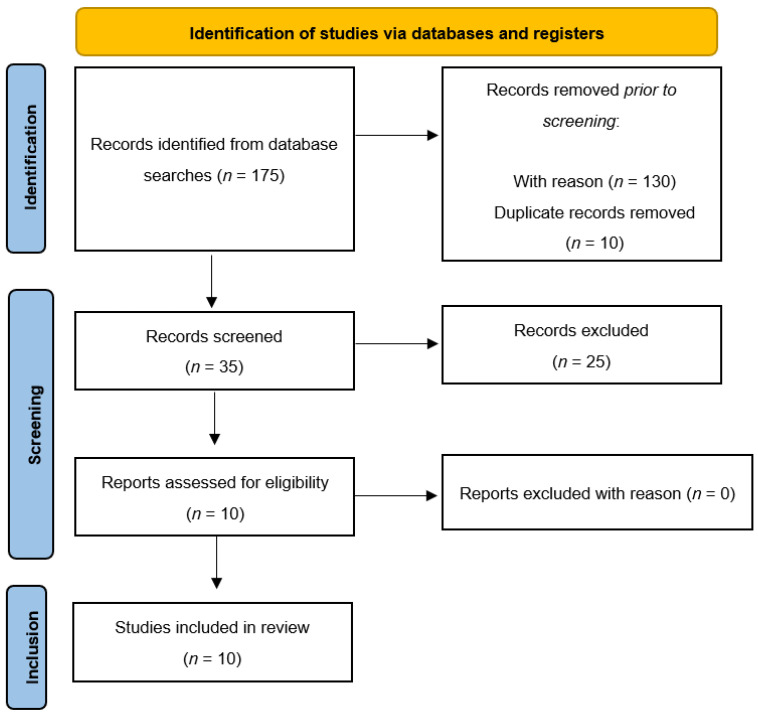
Flow chart of literature review.

**Table 1 jcm-11-06134-t001:** Excluded studies and the reasons for their exclusion.

Reason for Exclusion	Study Name
Article format (e.g., review)	Bhattacharya, A. et al. [3];
	Brytek-Matera, A. [22];
	Costa, C.B. et al. [23];
	Dell’Osso et al. [24];
	Dukay-Szabó, S. et al. [25];
	Dunn, T.M. et al. [26];
	Gortat, M. et al. [27];
	Goutaudier, N. et al. [28];
	Håman, L. et al. [29];
	Hyrnik, J. et al. [30];
	Kalra, S. et al. [31];
	McComb, S.E. et al. [32];
	Michalska, A. et al. [33];
	Niedzielski, A. et al. [34];
	Opitz, M.C. et al. [35];
	Strahler, J. [36];
	Strahler, J. et al. [37];
	Valente, M. et al. [38];
	Zagaria, A. et al. [39]
Sample characteristics: only specific population included	Bobonis Babilonia, M. et al. [40];
	Bóna, E. et al. [41];
	Domingues, R.B. et al. [42];
	Kinzl, J.F. et al. [43];
	Taştekin Ouyaba, A. et al. [44];
	Tremeling et al. [45]

## Data Availability

The study data are not available, due to ethical concerns. We must protect patient privacy and security and follow the ethical rules of our institutions and their restrictions on data sharing.

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
