# Peer review of "Orthorexia Nervosa, Eating Disorders, and Obsessive-Compulsive Disorder: A Selective Review of the Last Seven Years"

_jcm, 2022, doi:10.3390/jcm11206134_

Round 1
Reviewer 1 Report
This study is not fully considered novelty because some systematic reviews have published at same study focus in recent years.
At least, it is necessary to reconfirm the latest research trend of Orthorexia nervosa for publishing this study.
References (as the examples)
1) Orthorexia nervosa and eating disorder behaviors: A systematic review of the literature. Atchison AE, et al. Appetite. 2022 [doi: 10.1016/j.appet.2022.106134.]
2) Obsessed with Healthy Eating: A Systematic Review of Observational Studies Assessing Orthorexia Nervosa in Patients with Diabetes Mellitus.
Grammatikopoulou MG, et al. Nutrients. 2021 [doi: 10.3390/nu13113823.]
Author Response
Response: Thanks for your request of clarification. We have added the references above indicated. We have amended the text (introduction section) as follows:
“However, healthy eating represents the first-line prevention for various diseases and some studies have focused on orthorexia in specific clinical populations, for example patients with diabetes mellitus (Grammatikopoulou et al. 2021).”
“While in ON, weight loss is a common consequence of the highly restrictive diet, the alimentary behaviors aim at achieving feelings of purity and good health. Finally, Atchison and Zickgraf (2022), in a recent systematic review, found that ON symptoms are related to effort to lose or contain weight but not to body dissatisfaction or dysregulated eating. Therefore, according these authors, ON may represent a distinct ED.”
Finally, we tried to clarify the aim of our review (introduction section)
“Overall, there is limited systematic research on ON, its clinical significance, and its relationship with psychiatric disorders (e.g., EDs, OCD). The studies and reviews described so far have specifically considered the relationship between ON and eating disorders or that between ON and OCD. Thus, the present review aimed at summarizing the literature over the last 7 years on the clinical significance of ON and its relationship with both, EDs and OCD.”

Reviewer 2 Report
Really wonderful review with important implications. Thank you for your work.
A few suggestions. The summary of each study included in your analysis was brief, easy to read, and helpful. However, while you begin to describe some of significance of the findings of the studies in the discussion, you could provide a more in-depth report of what your report means. There are significant research and treatment implications of the cumulative findings of this review. For example, the paragraph that begins "the findings of other studies support the existence of a correlation...." is simply restating the findings of the review, rather than a discussion. Please consider going through each paragraph/point of your discussion section and providing the readers context of the significance of it and what it means for the future for researchers and clinicians. You should remove sentences that are simply restating findings unless they are tied directly into a discussion of that finding.
Importantly, this report does not assist the reader in understanding the significance of OCD or EDs in terms of physical and mental health symptoms. How does ON correlate with those aspects of why OCD or EDs are so serious and potentially life threatening? For example, what are the nutritional implications for ON in comparison to AN?
Thank you and I look forward to reading your finalized report.
Author Response
Reviewer #2
- Really wonderful review with important implications. Thank you for your work.
A few suggestions. The summary of each study included in your analysis was brief, easy to read, and helpful. However, while you begin to describe some of significance of the findings of the studies in the discussion, you could provide a more in-depth report of what your report means. There are significant research and treatment implications of the cumulative findings of this review. For example, the paragraph that begins "the findings of other studies support the existence of a correlation...." is simply restating the findings of the review, rather than a discussion. Please consider going through each paragraph/point of your discussion section and providing the readers context of the significance of it and what it means for the future for researchers and clinicians. You should remove sentences that are simply restating findings unless they are tied directly into a discussion of that finding.
Response: Thanks for your suggestion. We have modified discussion section. We have divided discussion in paragraph (Clinical significance of Orthorexia Nervosa; Orthorexia Nervosa and DSM-5 Eating Disorders; Orthorexia Nervosa and Obsessive-Compulsive Disorder) like the results section. Finally, we have modified the text as follow:
4.1 Clinical significance of Orthorexia Nervosa
“On this basis, ON may be considered a condition with a clinical significance and not simply a behavioral/lifestyle phenomenon. However, heterogeneous and culture-dependent definitions of what is considered normal eating behavior makes it complicated longitudinal studies on physical, psychological and social consequences and studies on pathological mechanism. Indeed, only a few researches have examined cognitive, emotional, or behavioral impairments that might be associated with ON. In addition, differentiating ON from already existing psychiatric is essential for its classification as a “new” mental illness with associate clinically relevant distress or limitations in important areas of life (e.g. social isolation).”
4.2 Orthorexia Nervosa and DSM-5 Eating Disorders
“In anorexia, a feeling of self-control (especially with respect to food intake), is a cru-cial clinical feature. Therefore, orthorexic eating behaviors might represent healthier ways of controlling food intake, shifting the focus from low-calorie foods to healthy foods, and resulting in a greater variety of “allowed” foods for recovering anorexic individuals and a lower risk of losing weight. This proposal by Barthels et al. [40] is supported by Se-gura-Garcia et al.’s study [11], which found that orthorexic eating behaviors were associ-ated with clinical improvement of disordered eating symptoms and a shift towards less severe forms of disturbed eating behaviors. Overall, since anorexia is difficult to treat and clinical changes need time, considering orthorexic eating behavior as a first step in psy-chotherapeutic approach might be a promising goal of treatment. However, further research is needed to assess orthorexic eating behaviors during treatment for anorexia, applying several follow-up measurements (i.e., not only self-report questionnaires) to investigate orthorexic and anorexic eating behaviors over time.”
4.3 Orthorexia Nervosa and Obsessive-Compulsive Disorder
“As regards the clinical relationship between ON and OCD, in line with other studies, Novara et al. showed a relationship between worry’s content and ON features.
Specifically, ON is characterized by overthinking about healthy food and food prep-aration rituals. These aspects resemble the obsessions and compulsive behaviors typical of the OCD. In addition, previous studies have shown that obsessive-compulsive symp-toms can be present in orthorexic individuals (Koven et al. 2013).
Vaccari et al. aimed at investigating the prevalence and intensity of ON symp-toms in OCD patients, in comparison to subjects with anxiety and depressive disorders and healthy controls. They found higher ON symptoms in OCD patients.
Despite this, we point out a crucial difference between ON and OCD. In individuals with ON, ritual food preparation is not aimed at reducing distress as in OCD compulsions.
Indeed, the repetitiveness and permanence of food-related thinking might suggest an excessive concern over the healthiness and preparation of food, rather than an obsession followed by a compulsion to neutralize it. Finally, individual with ON experiences his eating behavior as being in harmony or consistent with his ideal self-image. In addition,
We propose that considering this egosyntonicity is a crucial step in improving the diagnostic definition of orthorexia and the implementation of effective psychotherapeutic treatments. Even if obsessive-compulsive characteristics may be associated with or mediate orthorexic behaviors, these two conditions could be treated with different approaches.
For example, a psychoeducational intervention, focused on improving awareness of distress and the decrease in quality of life associated with orthorexic eating behaviors, could be useful in order to enhance compliance of individuals with ON with any other therapeutic intervention.”
- Importantly, this report does not assist the reader in understanding the significance of OCD or EDs in terms of physical and mental health symptoms. How does ON correlate with those aspects of why OCD or EDs are so serious and potentially life threatening? For example, what are the nutritional implications for ON in comparison to AN?
Response: Thanks for your request of clarification. We have amended the text (introduction section) as follows:
“In addition, ON is likely to lead to many of the same medical complications as EDs. For example, a lack of essential nutrients caused by restrictive eating can result in malnutrition, anemia, or an abnormally slow heart rate (Koven et al. 2015). Severe malnutrition may lead to gastrointestinal problems, hormonal imbalances, general fatigue, and a weakened immune system (Oberle et al. 2019)”
“Similar to the obsessions and compulsions of OCD patients, orthorexic thoughts and behaviors generate significant wasted time and limitations to social life resulting in high distress and worsening of quality of life (Koven et al. 2015).”

Reviewer 3 Report
The authors present a well organized narrative review of the literature regarding ON, EDs, and OCD. My only substantive suggestion is to modify the Discussion such that it serves as a commentary on the results, not a restatement of the results (as it reads now). This is the opportunity to synthesize and reflect on the results in each domain (ON as a distinct disorder; ON and OCD; ON as a lifestyle behavior) and to take on a broader perspective.
Author Response
Reviewer #3
- The authors present a well organized narrative review of the literature regarding ON, EDs, and OCD. My only substantive suggestion is to modify the Discussion such that it serves as a commentary on the results, not a restatement of the results (as it reads now). This is the opportunity to synthesize and reflect on the results in each domain (ON as a distinct disorder; ON and OCD; ON as a lifestyle behavior) and to take on a broader perspective.
Response: Thanks for your suggestion. We have modified discussion section. We have divided discussion in paragraph (Clinical significance of Orthorexia Nervosa; Orthorexia Nervosa and DSM-5 Eating Disorders; Orthorexia Nervosa and Obsessive-Compulsive Disorder) like the results section. Finally, we have modified the text as follow:
4.1 Clinical significance of Orthorexia Nervosa
“On this basis, ON may be considered a condition with a clinical significance and not simply a behavioral/lifestyle phenomenon. However, heterogeneous and culture-dependent definitions of what is considered normal eating behavior makes it complicated longitudinal studies on physical, psychological and social consequences and studies on pathological mechanism. Indeed, only a few researches have examined cognitive, emotional, or behavioral impairments that might be associated with ON. In addition, differentiating ON from already existing psychiatric is essential for its classification as a “new” mental illness with associate clinically relevant distress or limitations in important areas of life (e.g. social isolation).”
4.2 Orthorexia Nervosa and DSM-5 Eating Disorders
“In anorexia, a feeling of self-control (especially with respect to food intake), is a cru-cial clinical feature. Therefore, orthorexic eating behaviors might represent healthier ways of controlling food intake, shifting the focus from low-calorie foods to healthy foods, and resulting in a greater variety of “allowed” foods for recovering anorexic individuals and a lower risk of losing weight. This proposal by Barthels et al. [40] is supported by Se-gura-Garcia et al.’s study [11], which found that orthorexic eating behaviors were associ-ated with clinical improvement of disordered eating symptoms and a shift towards less severe forms of disturbed eating behaviors. Overall, since anorexia is difficult to treat and clinical changes need time, considering orthorexic eating behavior as a first step in psy-chotherapeutic approach might be a promising goal of treatment. However, further research is needed to assess orthorexic eating behaviors during treatment for anorexia, applying several follow-up measurements (i.e., not only self-report questionnaires) to investigate orthorexic and anorexic eating behaviors over time.”
4.3 Orthorexia Nervosa and Obsessive-Compulsive Disorder
“As regards the clinical relationship between ON and OCD, in line with other studies, Novara et al. showed a relationship between worry’s content and ON features.
Specifically, ON is characterized by overthinking about healthy food and food prep-aration rituals. These aspects resemble the obsessions and compulsive behaviors typical of the OCD. In addition, previous studies have shown that obsessive-compulsive symp-toms can be present in orthorexic individuals (Koven et al. 2013).
Vaccari et al. aimed at investigating the prevalence and intensity of ON symp-toms in OCD patients, in comparison to subjects with anxiety and depressive disorders and healthy controls. They found higher ON symptoms in OCD patients.
Despite this, we point out a crucial difference between ON and OCD. In individuals with ON, ritual food preparation is not aimed at reducing distress as in OCD compulsions.
Indeed, the repetitiveness and permanence of food-related thinking might suggest an excessive concern over the healthiness and preparation of food, rather than an obsession followed by a compulsion to neutralize it. Finally, individual with ON experiences his eating behavior as being in harmony or consistent with his ideal self-image. In addition,
We propose that considering this egosyntonicity is a crucial step in improving the diagnostic definition of orthorexia and the implementation of effective psychotherapeutic treatments. Even if obsessive-compulsive characteristics may be associated with or medi-ate orthorexic behaviors, these two conditions could be treated with different approaches.
For example, a psychoeducational intervention, focused on improving awareness of distress and the decrease in quality of life associated with orthorexic eating behaviors, could be useful in order to enhance compliance of individuals with ON with any other therapeutic intervention.”

Round 2
Reviewer 1 Report
The authors revised manuscript and explained for all reviewer's comment.
Author Response
Thank you for your feedback.
Reviewer 2 Report
-The abstract needs to be revised to be more clear with regards to the results (e.g. "a few studies" should be changed to X number of studies).
- Please define "pure foods", page 1 line 32
- I would include something in the introduction (likely in paragraph 3) outlining that dieting behavior is the number one predictor of onset of DSM5 categorized eating disorders and relate to your discussion of ON
- Please give more information about the ORTO-15. Give examples of the questions. How many questions are in it (I assume 15). What are the psychometric properties of the tool? You say that scores greater than 35 are pathological, but low scores are associated with high presence of orthorexic behaviors-- that seems to be contradictory? I thought this tool measured orthorexia, so if greater than 35 is pathologic for orthorexia then why would low scores be greatly associated with ON?
- Line 58-59 on page 2- please provide reference
- Line 61, page 2- how are the data on ON controversial? Please describe.
- Can you provide a distinction between the studies you included in the background/significance and the ones you included in analysis? Its not entirely clear to me what the difference is. I would probably focus the intro section more on what exactly is ON, AN, OCD, etc. and then the comparisons come from the lit review.
- In the intro you describe the ORTO-15 but then the first study you describe in the results uses a different measure (and other studies described below). Please include this measure in the intro. In the Strahler et al. paper, how did they measure obsessive compulsive behaviors?
- Line 210- what is BPNS-E?
-Line 222- what is the rBOT?
-Line 258- when describing all of the other studies, you simply state findings but here you are going into more of a discussion. Either summarize findings at the end of each study in this way or move this sentence to discussion.
-Line 265- what is the Y-BOCS? You need to spell each of these out the first time you use them.
-Line 334-336, the added sentence is run on/missing a word
- Line 350-The logic doesn't quite follow to say that higher orthorexic tendencies are associated with perfectionism and that is consistent with EDs. You're still talking about two different things: ON and EDs.
-Line 364- please cite this (crucial clinical feature)-- by what standards? self-control is not in the DSM5 so where are you getting that proposition.
- There are paragraphs in the OCD section of the discussion that are only one sentence long. Either expound on those thoughts or combine that with other paragraphs.
- self report data doesn't necessarily prevent quantitative analysis of the results. It is a limitation of analysis, for sure, but you can do analysis of self-report data.
I think this paper is really important work, but work needs to be done to clarify the difference between the intro and results, improve the results section in terms of description and robustness, and some careful thought needs to be applied to the analyses you are making in the discussion.
Author Response
[JCM] Manuscript ID: jcm-1873105
Response to Reviewers' Comments: We thank the reviewers for their scrutiny of the manuscript and insightful remarks, their very good feedback on our study was very encouraging. We hope to match their thoroughness and detail in our reply.
Please note that in the last version of the text we added bibliographical references. This altered the numbers of the lines you marked. Therefore, we have inserted our changes by referring to paragraph numbers.
Please note function that our replies are written in italics and that any changes to the text are marked up using the “Track Changes”
Response to Reviewer 2
- The abstract needs to be revised to be more clear with regards to the results (e.g. "a few studies" should be changed to X number of studies).
Response: Thanks for your suggestion. We have modified the text (abstract section) as follows:
“Finally, only two studies suggested a relationship between ON and OCD”.
- Please define "pure foods", page 1 line 32
Response: Thanks for your request of clarification. We have added the following sentence (introduction section)
“Notably, they exclude foods from their diets that they consider to be impure because they have pesticides or artificial substances and they worry in excess about the materials used in the food elaboration”.
- I would include something in the introduction (likely in paragraph 3) outlining that dieting behavior is the number one predictor of onset of DSM5 categorized eating disorders and relate to your discussion of ON.
Response: Thanks for your suggestion. We have amended the introduction section as follows:
“As concerns the relationship between ON and EDs, Dell’Osso et al. (2018) proposed that ON and EDs (e.g., anorexia nervosa) share abnormal eating attitudes with dieting behavior and poor insight. Regarding this, a previous study ( Stice et al, 2017) showed that dieting behavior is among the factors that increase the risk for EDs. Other factors are thin ideal, body dissatisfaction and unhealthy weight control behaviors. Finally, ON is likely to lead to many of the same medical complications as EDs. For example, a lack of essential nutrients caused by restrictive eating can result in malnutrition, anemia, or an abnormally slow heart rate (Koven, 2015). Severe malnutrition may lead to gastrointestinal problems, hormonal imbalances, general fatigue, and a weakened immune system (Oberle, 2018).
- Please give more information about the ORTO-15. Give examples of the questions. How many questions are in it (I assume 15). What are the psychometric properties of the tool? You say that scores greater than 35 are pathological, but low scores are associated with high presence of orthorexic behaviors-- that seems to be contradictory? I thought this tool measured orthorexia, so if greater than 35 is pathologic for orthorexia then why would low scores be greatly associated with ON?
Response: Thanks for your request of clarification. We confirm that low scores in Orto-15 are associated with high presence of orthorexic behaviors (see introduction: “Total scores < 35 are considered pathological”). Indeed, according to Donini et al. 2015, “Answers that indicated orthorexia were given a score of “1”, while the “healthier” ones had a score of “4”. The sum of the scores was the final score of the test.
We have added (introduction section) some more information about ORTO-15:
“The ORTO-15 is a self-report questionnaire and made up of 15 multiple choice items (e.g. When eating, do you pay attention to the calories of the food?; Are you willing to spend more money to have healthier food; In the last 3 months, did the thought of food worry you). Total scores < 35 are considered pathological, and low scores, more generally, are significantly associated with a high presence of orthorexic behaviors. According to Donini et al. (2005), the overall accuracy of the ORTHO-15 test as a test for the diagnosis of ON was found to be 0.696 (95% CI: 0.585-0.807).However, different versions of the instrument (e.g., ORTO-14, ORTO-R) contain different numbers of items.
- Line 58-59 on page 2- please provide reference.
Response: Thanks for your request of clarification. We have added bibliographic references as follows (introduction section)
“Previous research (Chaki, Pal, & Bandyopadhyay, 2013; Mathieu, 2005; Scarff, J. R. (2017) has revealed that the clinical characteristics of ON are similar to those of two psychopathological disorders: eating disorders (EDs) and obsessive-compulsive disorder (OCD).
- Line 61, page 2- how are the data on ON controversial? Please describe.
Response: Thanks for your request of clarification. We noticed that the sentence “However, only scarce data are available on ON, and even these data are controversial” is not adequate in this paragraph and we have removed it.
- Can you provide a distinction between the studies you included in the background/significance and the ones you included in analysis? Its not entirely clear to me what the difference is.
Response: Thanks for your request of clarification. In the background/introduction section we included reviews, commentary and meta-analyses on orthorexia nervosa but also studies that did not fit our inclusion criteria but focused on orthorexia nervosa. In the analysis section we included only studies that investigated the clinical significance of ON and its relationship with EDs and OCD. (see 2.2 paragraph and Table 1).
- In the intro you describe the ORTO-15 but then the first study you describe in the results uses a different measure (and other studies described below). Please include this measure in the intro.
Response: Thanks for your request of clarification. We have amended introduction section as follows:
“Other instruments are also used for the evaluation of ON. For example, The Düsseldorf Orthorexie Scale (DOS) is a 10-item self-reported questionnaire and measures orthorexic eating behavior. A four-point Likert-scale from “this applies to me” (4 points) to “this does not apply to me” (1 point) is used. Higher points indicate more pronounced orthorexic behavior. As a cut-off score to indicate presence of ON, a score ≥30 is used. DOS has acceptable internal consistency (a=.84) (Barthels 2015). Finally,the Revised Bratman’s Orthorexia Test [rBOT] Haeberle -Savard, 2015 ) is an eight-item modified version of the Bratman Orthorexia Test [BOT] (Bratman & Knight, 2000 ). Scores are summed with a possible range of 0 -16. The rBOT has good internal consistency ( a = .85) (Barthels et al. 2020)”
- In the Strahler et al. paper, how did they measure obsessive compulsive behaviors?
Response: Thanks for your request of clarification. We have amended the text (paragraph 3.1) as follows:
“Finally, obsessive and compulsive behaviors (measured using Yale Brown Obsessive-Compulsive Scale and Symptom Checklist [ Y-BOCS]) were detected in 30% of the ON subjects.”
- Line 210- what is BPNS-E?
Response: Thank you for your request of clarification. We have modified as follows (see in Paragraph 3.2):
“Finally, regarding the fulfilment of basic psychological needs, the ANO group scored significantly higher on the Basic Psychological Needs Scale [BPNS-E] competence (p < .017) and autonomy (p < .017) subscales, relative to the AN group.”
- Line 222- what is the rBOT?
Response: Thank you for your request of clarification. We have modified as follows (see in Paragraph 3.2):
“Bartel et al. (2020) investigated whether ON bears a stronger relationship with EDs or OCD in a sample of 512 young adults, using the the Revised Bratman’s Orthorexia Test [rBOT], the Eating Disorder Examination Questionnaire (EDE-Q), the Obsessive-Compulsive Inventory Revised (OCI-R), the Food Choice Questionnaire, and the Frost Multidimensional Perfectionism Scale (FMPS).”
- Line 258- when describing all of the other studies, you simply state findings but here you are going into more of a discussion. Either summarize findings at the end of each study in this way or move this sentence to discussion.
Response: Thanks for your suggestion. We have moved this sentence to discussion.
- Line 265- what is the Y-BOCS? You need to spell each of these out the first time you use them
Response: Thanks you for your request of clarification. We have already inserted the explanation in the text (paragraph 3.1) as follows:
Finally, obsessive and compulsive behaviors (measured using Yale Brown Obsessive-Compulsive Scale and Symptom Checklist [ Y-BOCS]) were detected in 30% of the ON subjects.
- Line 334-336, the added sentence is run on/missing a word
Response: Thanks for your suggestion. We have modified discussion section (see paragraph 4.1) as follows:
“However, heterogeneous and culture-dependent definitions of what is considered normal eating behavior makes it complicated longitudinal studies on physical, psychological and social consequences of ON.”
- Line 350: The logic doesn't quite follow to say that higher orthorexic tendencies are associated with perfectionism and that is consistent with EDs. You're still talking about two different things: ON and EDs.
Response: Thanks for your request of clarification.
Paragraph 4.1: we have cited Barnes and Caltabiano (2016) that showed the relationship between higher orthorexic tendencies and higher perfectionism. Contemporary, we have cited the findings of other studies (Brown et al. 2012; Bardone-Cone et al. 2007) showing that perfectionism is implicated in the development and maintenance of EDs. Aim of this paragraph was to try to describe the psychological mechanisms that orthorexia and EDs could share on the basis of some cited research.
For this purpose, we have modified the discussion as follows:
“Barnes and Caltabiano (2016) showed that higher orthorexic tendencies significantly correlated with higher perfectionism (i.e., self-oriented, others-oriented, socially prescribed), overweight preoccupation, and appearance orientation. These findings are consistent with the results of other studies finding that perfectionism is implicated in the development and maintenance of EDs [52,53]. Also, greater appearance orientation and overweight preoccupation has been observed among anorexic and bulimic individuals [54]. Overall, this may suggest that orthorexia nervosa shares similarities with EDs with regards to perfectionism, overweight preoccupation and appearance orientation. In addition, in Barnes and Caltabiano’s study, history of an ED, appearance orientation, and overweight preoccupation emerged as the most important predictors of ON. This may suggest that an exaggerated focus on appearance and a fear of becoming overweight might be underlying motivations for ON individuals’ preoccupation with a healthy diet, rather than a fixation on health, per se.”
- Line 364- please cite this (crucial clinical feature)-by what standards? self-control is not in the DSM5 so where are you getting that proposition.
Response: Thanks for your request of clarification. We have modified discussion section (Paragraph 4.2) as follows:
“In anorexia, according to Fairburn et al. (1999), a feeling of self-control (especially with respect to food intake) is a relevant clinical feature.”
- There are paragraphs in the OCD section of the discussion that are only one sentence long. Either expound on those thoughts or combine that with other paragraphs.
Response: Thanks for request of clarification: We have modified discussion (4.3 paragraph) as follows:
“As regards the clinical relationship between ON and OCD, previous studies (Cena et al. 2019; Bratman et al. 1997; Koven et al. 2013) have shown that obsessive-compulsive symptoms can be present in orthorexic individuals (Koven et al. 2013). In our review, Novara et al. (2021) focused on the relationship between worry’s content and ON clinical features. Specifically, ON is characterized by overthinking about healthy food and food preparation rituals. These aspects resemble the obsessions and compulsive behaviors typical of the OCD. Similarly, Vaccari et al. (2021) aimed at investigating the prevalence and intensity of ON symptoms in OCD patients, in comparison to subjects with anxiety and depressive disorders and healthy controls. They found higher ON symptoms in OCD patients.Overall, these results seems to support the idea that a correlation exists not only between ON and EDs, but also between ON and OCD.
Despite this, we point out a crucial difference between ON and OCD. In individuals with ON, ritual food preparation is not aimed at reducing distress as in OCD compulsions.
Indeed, the repetitiveness and permanence of food-related thinking might suggest an excessive concern over the healthiness and preparation of food, rather than an obsession followed by a compulsion to neutralize it. Finally, individual with ON experiences his eating behavior as being in harmony or consistent with his ideal self-image.
We propose that considering this egosyntonicity is a crucial target in improving the diagnostic definition of orthorexia and the implementation of effective psychotherapeutic treatments. Even if obsessive-compulsive characteristics may be associated with or mediate orthorexic behaviors, these two conditions could be treated with different approaches.
For example, a psychoeducational intervention, focused on improving awareness of distress and the decrease in quality of life associated with orthorexic eating behaviors, could be useful in order to enhance compliance of individuals with ON with any other therapeutic intervention.”
- Self report data doesn't necessarily prevent quantitative analysis of the results. It is a limitation of analysis, for sure, but you can do analysis of self-report data.
Response: Thanks for your suggestion. We have modified limitations sections as follows:
“Some limitations should be considered in our review. Firstly, the included studies featured a variety of study designs and ON measures. In addition, the review included both general population and clinical population studies. This limits the interpretability of the findings.”
